# Towards Improving Exploration through Sibling Augmented GFlowNets

**Kanika Madan[1], Alex Lamb[2], Emmanuel Bengio[3], Glen Berseth[1], Yoshua Bengio[1]**

## Abstract

Exploration is a key factor for the success of an active learning agent, especially when dealing with sparse extrinsic terminal rewards and long trajectories. We introduce Sibling Augmented Generative Flow Networks (SA-GFN), a novel framework designed to enhance exploration and training efficiency of Generative Flow Networks (GFlowNets). SA-GFN uses a decoupled dual network architecture, comprising of a main Behavior Network and an exploratory Sibling Network, to enable a diverse exploration of the underlying distribution using intrinsic rewards. Inspired by the ideas on exploration from reinforcement learning, SA-GFN provides a general-purpose exploration and learning paradigm that integrates with multiple GFlowNet training objectives and is especially helpful for exploration over a wide range of sparse or low reward distributions and task structures. An extensive set of experiments across a diverse range of tasks, reward structures and trajectory lengths, along with a thorough set of ablations, demonstrate the superior performance of SA-GFN in terms of exploration efficacy and convergence speed as compared to the existing methods. In addition, SA-GFN's versatility and compatibility with different GFlowNet training objectives and intrinsic reward methods underscores its broad applicability in various problem domains.

## 1 Introduction

Exploration is a fundamental aspect of a learning agent that is actively interacting with its environment and learning from the experience collected. A sufficient amount of exploration also leads to a better generalization, enabling the agent to learn from a diverse set of experiences collected from different regions of the underlying state space. In reinforcement learning, this is commonly referred to as the exploration-exploitation trade-off (Sutton & Barto, 2018). An agent that only exploits and doesn't explore can get stuck and is unable to collect novel experiences to further improve its policy. In the absence of any intermediate reward signals, the learning agent might need to traverse several low or zero reward regions in order to discover the ones with high-rewards, making exploration a non-trivial task, especially when faced with a large state space, sparse terminal reward signals and complex reward distributions.

Manually designing dense rewards has been successful in many areas of robotics and games (Mnih et al., 2015; Baker et al., 2019; Hafner et al., 2019), but this approach is not easy to design and scale. Unstructured exploration based methods, such as $\epsilon$-greedy or randomized probability matching, are not able to effectively leverage any structure and hence are not efficient beyond simple settings. Intrinsic motivation methods have been successfully used (Pathak et al., 2017; Burda et al., 2018; Badia et al., 2020; Zhang et al., 2021) to guide exploration over a range of problems, and are based on generating intrinsic rewards, or rewards internal to an agent, to encourage visiting novel states. Random Network Distillation (RND) (Burda et al., 2018) is a commonly used intrinsic reward method that is easy to use and scale, and generates intrinsic rewards defined in terms of an evolving distance metric based on the fixed and learnt representations of a given state.

At the same time, for many scenarios such as biological sequence design (Jain et al., 2023) and molecule generation (Bengio et al., 2021a), diversity can be an equally important aspect for the learnt policy. While reinforcement learning (RL) methods are based on learning a reward-maximizing

---

[01] Mila – Québec AI Institute, Université de Montréal, [2] Microsoft Research, [3] Valence Labs.
Corresponding author: `madankanika.s@gmail.com`

policy that can sample from the highest mode of the distribution, generating diverse high-reward solutions might be what we actually care about.

Generative Flow Networks, (GFlowNets; Bengio et al., 2021a), are amortized variational inference algorithms that learn a stochastic forward policy $\pi(a|s)$ to generate an object $x$ in proportion to its (positive) reward $R(x)$. Unlike RL, GFlowNets can generate diverse high-rewarding candidates. However, similar to RL, the training framework of GFlowNets is based on actively sampling trajectories from the forward policy and learning from the corresponding terminal rewards. Therefore, the success of a GFlowNet policy also depends on achieving a good exploration-exploitation trade-off to ensure that the policy has been able to discover high-rewarding modes of the target distribution.

Some previous works, such as Malkin et al. (2022); Deleu et al. (2022); Madan et al. (2023); Pan et al. (2023) have addressed improving credit assignment in GFlowNets, and Pan et al. (2022); Rector-Brooks et al. (2023); Lau et al. (2023), have looked into exploration. However, the range of exploration tasks that have been considered so far are limited to smaller objects to be generated, relatively easier reward settings, and extremely sparse reward settings.

In this work, we aim to reduce this gap by proposing a general framework dubbed **Sibling Augmented Generative Flow Networks** or **SA-GFN**, that can not only explore better, but can also learn faster over a wide range of tasks that are not just limited to easy reward or extremely sparse zero-reward settings previously considered. We also expand the set of previous exploration benchmarks to include non-zero, yet difficult to explore, sparse reward structures, as well as challenging sequence design problems for both short and long range trajectories. To our knowledge, this is the first extensive evaluation focused on the exploration aspect of GFlowNets.

The main contributions of our work are the following:

1. We propose a novel general-purpose framework, called *Sibling Augmented Generative Flow Networks* or *SA-GFN*, that achieves much better exploration and training of GFlowNets using intrinsic rewards and a disentangled dual network architecture.

2. We show the generality of SA-GFN, which is in principle compatible with any GFlowNet objective and any intrinsic reward method, and hence is simple to work with.

Through an extensive set of experiments and ablations over a wide range of tasks and difficulty levels, we establish the exploration benefits of SA-GFN compared to the previously introduced methods.

## 2 RELATED WORK

### 2.1 INTRINSIC MOTIVATION BASED EXPLORATION METHODS

Intrinsic motivation based methods encourage exploration by providing additional intrinsic rewards, i.e. rewards internal to the agent, for visiting unexplored regions of the state space.

**Prediction Error based methods:** Predictions from a learnt world dynamics model are compared against the ground truth, and higher prediction errors provide higher intrinsic rewards (Achiam & Sastry, 2017; Li et al., 2019), encouraging the policy to explore unknown regions of the distribution.

**Novelty based methods:** The intrinsic reward is defined in terms of an evolving distance metric between the learnt embedding and a target embedding of the visited states. Count-based methods are early examples of this class of methods (Bellemare et al., 2016; Ostrovski et al., 2017; Tang et al., 2016), but these do not scale well. A more scalable variation is Random Network Distillation or RND (Burda et al., 2018) in which representations from a randomly initialized target network are distilled into a learnt network, providing an evolving distance metric; several works (Pathak et al., 2017; Badia et al., 2020; Zhang et al., 2020; 2021) have built on this framework.

### 2.2 EXPLORATION FOR TRAINING GFLOWNETS

Similar to reinforcement learning, GFlowNets also face the problem of exploration when learning over long trajectories, high-dimensional spaces, sparse rewards and non-uniform reward settings

with difficult-to-explore reward structures. Connections between RL and GFlowNets have been made in **?**. Although intrinsic rewards were incorporated into the training objectives of GFlowNets in Pan et al. (2022), only extremely sparse settings with zero rewards were considered, and we empirically found it to not be able to explore well to more general, difficult and low (not necessarily zero) reward settings (see Section 5). Rector-Brooks et al. (2023) used a variation of Thompson Sampling using an ensemble of K forward policies to estimate uncertainty (Osband et al., 2016; 2018) to guide exploration. However, the size of such ensemble methods is a sensitive parameter and these methods tend to not scale well. Lau et al. (2023) proposed to use a moving average copy of the online network to be used as a sampler to stabilize training and improve exploration, but only considered settings where exploration is not as much of an issue.

## 3 BACKGROUND

### 3.1 GFLOWNETS

Consider a directed acyclic graph (DAG), $G = (S, \mathbb{A})$, such that $S$ represents the set of nodes and $\mathbb{A}$ represents the set of edges $(s \rightarrow s') \in \mathbb{A}$. Given an edge $(s \rightarrow s')$, $s$ is called the *parent* of $s'$ and $s'$ is called the *child* of $s$. The DAG is called pointed because it has a unique root node with no parents, also called *source* node or initial state $s_0$. *Terminal* states do not have any children and belong to the set $\mathcal{X}$ of objects that the GFlowNet policy could constructively sample through a sequence of actions. A *trajectory* $\tau = (s_m, s_{m+1}, \ldots, s_{n-1}, s_n)$ is formed by a sequence of actions $(s_i \rightarrow s_{i+1})$ and is called *complete* if $s_m = s_0$ and $s_n$ is a terminal state.

GFlowNets learn a stochastic forward policy $\pi$, that can take a sequence of actions to sequentially generate an object $x$ in proportion to its reward $R(x)$. Each such action taken by the GFlowNet modifies the state, for example by adding an element to the partially constructed object generated so far, until the policy decides to stop and a terminal object $x$ is generated, after which a corresponding reward $R(x)$ is provided by the environment. We emphasize that there are only terminal rewards provided at the end of the trajectory.

### 3.2 GFLOWNETS TRAINING

Many training objectives have been defined for GFlowNets, such as Flow Matching objective (Bengio et al., 2021a), Detailed Balance objective (Bengio et al., 2021b), Trajectory Balance objective (Malkin et al., 2022) and SubTB($\lambda$) objective (Madan et al., 2023), and these operate on the level of the state, edge, full length (complete) trajectories and sub-trajectories of any lengths, respectively. These training objectives are obtained by setting up a set of flow-matching constraints with the property that when all these constraints are satisfied, the GFlowNet sampling policy has the desired property that generates terminal states with probability proportional to the given reward function. Each constraint can be turned into a loss, typically by taking the square of the logarithm of the ratio of the right-hand side to the left-hand side of the equality constraint. Each loss term thus corresponds to an amount of constraint violation. Training consists in sampling trajectories and measuring these constraint violations (the loss) and its gradient on the parameters of interest. Furthermore, they all enable off-policy training, i.e., these training trajectories can be sampled from any full-support distribution. However, some distributions will yield faster convergence because they focus on areas where the rewards are higher and where the current policy has not yet explored.

The Flow Matching (FM) (Bengio et al., 2021a) objective parameterizes GFlowNets through edge flows $F(s \rightarrow s'; \theta)$ on states $s$. The Detailed Balance (DB) (Bengio et al., 2021b) and the SubTB($\lambda$) (Madan et al., 2023) objectives paramaterize the state flow $F(s; \theta)$, forward policy $P_F(s'|s; \theta)$, and backward policy $P_B(s|s'; \theta)$ on actions $s \rightarrow s'$ to define a GFlowNet. The Trajectory Balance (TB) (Malkin et al., 2022) objective works with complete trajectories, and parameterizes the GFlowNet through an initial state flow $Z_\theta$, and forward and backward policies $P_F(s|s; \theta)$, $P_B(s|s'; \theta)$ respectively. The flow-matching constraints represented by these parameterized quantities are converted into a loss function by equating the left and right hand sides of the constraint equations as a squared loss. The flow matching equation for the Trajectory Balance loss is shown in Eq. 1.

$$Z_\theta \prod_{i=0}^{n-1} P_{F_\theta}(s_{i+1}|s_i) = R(s_n) \prod_{i=0}^{n-1} P_{B_\theta}(s_i|s_{i+1}). \tag{1}$$

# 4 SIBLING AUGMENTED GENERATIVE FLOW NETWORKS (SA-GFN)

We propose a flexible GFlowNet learning framework, dubbed **Sibling Augmented Generative Flow Networks** or **SA-GFN**, that incorporates intrinsic rewards in a simple manner and leverages the off-policy learning capabilities of GFlowNets to learn from exploratory data and better match the underlying energy function. Specifically, SA-GFN uses Random Network Distillation (RND) (Burda et al., 2018) as intrinsic rewards, and has a disentangled dual network architecture in which an exploratory sibling policy provides exploratory data to train the main behavior policy. By disentangling exploration from learning, SA-GFN enables simultaneous efficient exploration and effective learning.

RND-based intrinsic rewards are defined such that states with higher novelty reap larger intrinsic rewards. However, directly incorporating a continuously evolving reward into the training objective of GFlowNets makes the target reward distribution a moving target; this can negatively impact learning. We empirically show that by disentangling the exploration network (trained using the RND based intrinsic rewards) from the main behavior network (trained using rewards from the true distribution), the proposed SA-GFN achieves a better exploration and learning of the underlying distribution.

## 4.1 SA-GFN ARCHITECTURE

Sibling Augmented Generative Flow Networks (SA-GFN) adopts a decoupled architecture consisting of two separate networks: a Sibling Network and a Behavior Network. The Behavior Network is the main GFlowNet that aims to learn the true target reward distribution, while the Sibling Network is an exploratory policy network that uses intrinsic rewards to explore the space. The exploratory data collected by the Sibling Network is relabeled with the true rewards (Andrychowicz et al., 2017), and is combined with the on-policy data collected by the Behavior Network to train its forward policy. The off-policy learning capabilities of GFlowNets and relabeling of trajectories frees the Behavior Network from the task of modeling the continuously evolving intrinsic rewards, enabling SA-GFN to achieve an efficient exploration.

The main components of the proposed SA-GFN architecture are as follows.

Figure 1: Sibling Augmented Generative Flow Networks (SA-GFN) has a decoupled architecture, consisting of (a) a main Behavior Network with policy $\pi^{BN}$ and (b) an exploratory Sibling Network with policy $\pi^{SN}$. Exploratory trajectories, $\tau^{SN}$, sampled using intrinsic rewards from policy $\pi^{SN}$ are (a) used to update the Sibling Network and (b) relabeled with true rewards, $\tau_{SN}^{BN}$ and combined with $\tau^{SN}$ to update the Behavior Network.

**Intrinsic Reward Network:** This module generates intrinsic rewards for a given set of sampled trajectories. In SA-GFN, we use Random Network Distillation (RND) (Burda et al., 2018) to compute these intrinsic rewards in which a randomly generated target network is distilled into a learnt predictor network. As a consequence, novel states, or those that are difficult to predict, tend to reap higher intrinsic rewards. In principle, any other intrinsic reward as well as extensions of RND can be used and we provide the evidence for this in Section 10.3.

**Sibling Network:** The Sibling Network uses intrinsic reward based exploration to sample trajectories $\tau^{SN}$ according to the reward $r_\tau^{SN}$ defined in Eq. 2, where $\beta_{e_{SN}}, \beta_i$ and $\beta_{SN}$ are the hyperparameters for the reward exponents, and $r_t^i, r_t^e$ represent the intrinsic and extrinsic rewards, respectively, at time $t$. Any of the GFlowNet training objectives discussed in Section 3.2 can be used to train the Sibling Network using $(\tau^{SN}, r_\tau^{SN})$. We use the TB loss in our experiments, and provide extension to the other GFlowNet objectives in Section 10.4.

$$r_\tau^{SN} = ((r_t^e)^{\beta_{e_{SN}}} + (\sum_t r_t^i)^{\beta_i})^{\beta_{SN}}. \tag{2}$$

**Behavior Network:** The Behavior Network is the main GFlowNet that learns to sample according to a given extrinsic reward function. The Behavior GFlowNet learns from two sets of trajectories: (a) on-policy trajectories and corresponding rewards $(\tau^{BN}, r_\tau^{BN})$ generated by the forward policy of the Behavior Network, and (b) exploratory trajectories generated by the Sibling Network with relabeled rewards, denoted by $(\tau_{SN}^{BN}, r_t^e)$. Since GFlowNets can be trained off-policy, the two sets of trajectories are combined $(\{\tau^{BN} \cup \tau_{SN}^{BN}\}, \{r_\tau^{BN} \bigcup r_t^e\})$ and are used together to train the Behavior Network.

By decoupling the training of the main Behavior Network from that of the exploratory Sibling Network, any optimization instabilities arising from a continuously changing intrinsic rewards do not have a negative impact on the training of the main Behavior Network. Instead, learning from the exploratory data generated by the Sibling Network allows a very efficient exploration of the target distribution, and an extensive set of experiments over a diverse set of reward settings corroborate this in Section 5. Moreover, this decoupling makes the SA-GFN framework general enough that any of the GFlowNet training objectives and any of the variants of intrinsic rewards can be seamlessly incorporated, as empirically shown in Sections 10.4 and 10.3.

### 4.2 SA-GFN TRAINING OBJECTIVES

Any of the GFlowNet training objectives from Section 3.2 can be used to train the Behavior and Sibling networks of the SA-GFN architecture. Trajectories generated

---

**Algorithm 1:** Sibling Augmented Generative Flow Networks (SA-GFN)

**Require:** Sibling Network or SN: $P_F^{SN}(s'|s), P_B^{SN}(s|s'), Z^{SN}$; Behavior Network or BN: $P_F^{BN}(s'|s), P_B^{BN}(s|s'), Z^{BN}$; random target network: $\bar{\phi}$; predictor network: $\phi$;

1: **Input**
2:   $\beta_{e_{BN}}, \beta_{e_{SN}}$: reward exponents for extrinsic rewards, $r_t^e$, for Behavior Network and Sibling Network ;
3:   $\beta_i$: reward exponent for intrinsic rewards $r_t^i$;
4:   $\beta_{SN}, \beta_{BN}$: reward exponents for final rewards of Sibling Network and Behavior Network;
5: **for** each training iteration **do**
6:     Collect trajectories $\tau^{SN}$ using the forward policy $P_F^{SN}(s'|s)$ of the Sibling Network
7:     Compute reward $r_\tau^{SN}$ for Sibling Network using Eq. 2.
8:     *// Update the Sibling Network*
    Compute $\mathcal{L}^{SN}(\tau^{SN}, r_\tau^{SN})$ using Eq. 1 and update the Sibling Network $P_F^{SN}(s'|s), P_B^{SN}(s|s'), Z^{SN}$.
9:     Collect trajectories $\tau^{BN}$ & extrinsic rewards $r_\tau^{BN}$ using the forward policy $P_F^{BN}(s'|s)$ of the Behavior Network.
10:    Relabel: $(\tau_{SN}^{BN}, r_\tau^{SN}) \leftarrow (\tau^{SN}, r_t^e)$
11:    Update: $\tau^{BN} \leftarrow \{\tau^{BN} \cup \tau_{SN}^{BN}\}$
12:    Update: $r_\tau^{BN} \leftarrow \{r_\tau^{BN} \cup r_\tau^{SN}\}$
13:    *// Update the Behavior Network*
    Compute $\mathcal{L}^{BN}(\tau^{BN}, r_\tau^{BN})$ using Eq. 1 and update the Behavior Network $P_F^{BN}(s'|s), P_B^{BN}(s|s'), Z^{BN}$
14:    *// Update the Intrinsic Reward Network*
    Update the intrinsic reward network $\phi$ using RND loss $||\bar{\phi}(s) - \phi(s)||_2$
15: **end for**

---

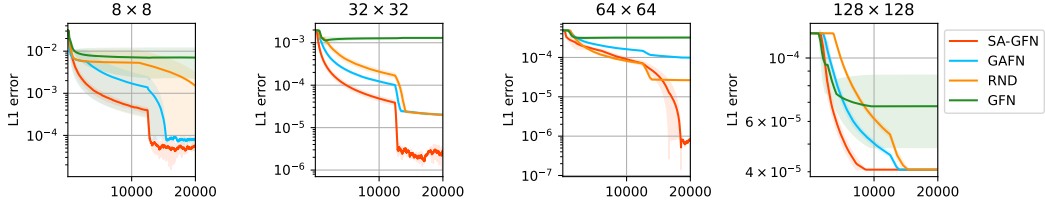

Figure 2: For a Sparse HyperGrid with zero-rewards, SA-GFN outperforms all other baselines, highlighting faster convergence, efficient exploration, and better learning of the true reward distribution.

by the Sibling Network are relabeled with true extrinsic rewards (Andrychowicz et al., 2017) to train the Behavior Network using off-policy learning. To encourage visiting novel states, the Intrinsic Reward Network is based on, but not limited to (as shown in 10.3), RND (Burda et al., 2018). Full algorithm is laid out in Algo 1 and Figure 1.

### 4.3 HYPOTHESIZED BENEFITS OF SA-GFN

In addition to improving exploration, we hypothesize the following other benefits of SA-GFN.

**Better Training**    Learning only from a terminal rewards can severely limit the training efficiency of an agent. Having intermediate rewards, such as intrinsic rewards, can induce a more efficient exploration. However, since GFlowNets learn to match a fixed target reward distribution, using non-stationary rewards during training creates moving targets, leading to inefficiencies during training. The proposed SA-GFN decouples exploration from training by using a separate Sibling Network to generate exploratory trajectories that are relabeled and used to train the main Behavior Network. The target reward distribution of the Behavior Network thus stays unchanged, while experience from unexplored regions of the distribution is being constantly fed, allowing SAGFN to explore better as compared to the single network variants, as evident through results shown in Section 5.

**Flexible & Expandable Architecture**    SA-GFN allows using any of the GFlowNet training objectives and architectures to train the models, shown in 10.4. Moreover, other intrinsic reward methods, including extensions of RND, can be used for the Intrinsic Reward Module of the Sibling Network, as shown in 10.3. SA-GFN also allows us to use other techniques to improve training, such as a replay buffer, multiple heads or ensemble  (Rector-Brooks et al., 2023), other exploration focused variants of GFlowNets, reward exponents and tempered policies (Kim et al., 2024).

## 5   EXPERIMENTS

To evaluate SA-GFN against other baselines, we address the following research questions:

1. *Number of discovered modes:*  We track the number of modes learnt by each method over a diverse range of task structures and reward settings; Section 5.1 and 5.2.

2. *Learning of the true reward distribution:*  We measure the $L1$ error between the true reward distribution and the learnt empirical distribution for each method, Section 5.1. We also visualize the learnt empirical distributions at the end of the training to compare against the true reward; Section 10.7.

3. *Efficient exploration under a variety of tasks and reward structures:*  We extend the range of tasks to cover a wide range of difficult exploration settings and difficult to explore reward structures; Section 5.1, 5.2 and 5.3.

4. *Robustness to trajectory length and size of the action space:*  We test all methods over a large range of trajectory lengths and varying dimensionalities of the action space for multiple domains and task structures; Section 5.1, 5.2.

To evaluate on a wide range of exploration tasks, we conducted experiments on the following four domains, which to our knowledge covers the widest range of exploration settings considered so far: (a) *Sparse Zero-Reward HyperGrid:* Hypergrid with zero-reward regions; (Section 5.1.1), (b) *Sparse*

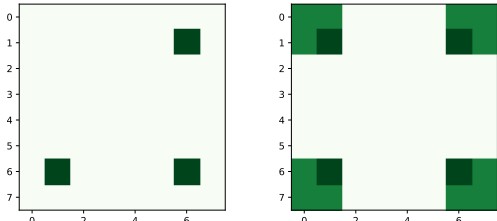

Figure 3: Sparse Rewards: [**Left**] zero rewards [**Right**] very low non-zero rewards, and high-reward corners.

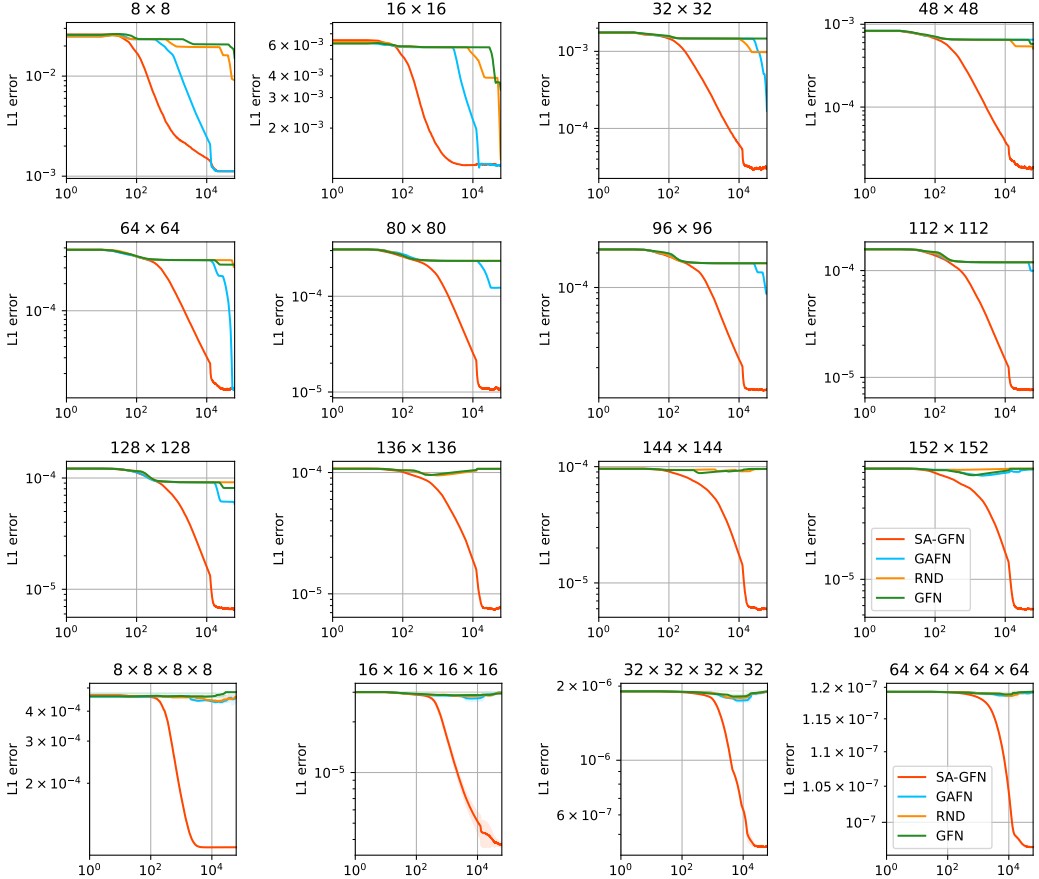

Figure 4: For the Sparse HyperGrid with a low-reward, yet difficult to explore, setting, SA-GFN outperforms all other baselines methods, highlighting its benefits in terms of faster convergence, efficient exploration, and a better learning of the underlying true reward distribution.

*Low-Reward HyperGrid:* HyperGrid with regions of very low, non-zero, difficult to explore rewards (Section 5.1.2), (c) *Bit Sequence Task:* from Malkin et al. (2022) involving auto-regressive generation of sequences of varying lengths and vocabulary sizes (Section 5.2), and (d) *Small Molecule Generation:* from (Bengio et al., 2021a), involving generation of sEH protein binders (Section 5.3).

## 5.1 HYPERGRID

The HyperGrid environment is a $d$-dimensional grid of size $H \times H \times \cdots \times H$ with an initial state $s_0$: $(0, 0, \ldots, 0)$. To ensure a DAG structure, each action increments one of the $d$ coordinates by one without leaving the grid, with an additional stop action to end the trajectory.

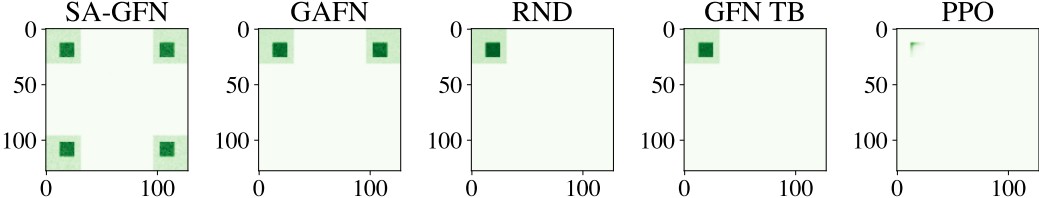

Figure 5: SA-GFN learns all the four modes of the distribution while other methods cannot, showing their poor exploration properties in a Sparse HyperGrid with difficult to explore rewards setting.

In order to evaluate over a broad spectrum of hard exploration problems, we control exploration difficulty through the following:

***Grid size:*** Larger grids necessitate traversing longer trajectories through very low reward or sparse reward spaces in order to find high rewarding modes. Thus, larger the grid, harder the exploration.

***Grid Dimensionality:*** A HyperGrid spanning multiple dimensions can be difficult to explore, even for shorter trajectories. For example, a 2-dimensional grid with $H = 100$ and a 4-dimensional grid with $H = 10$ have the same number of terminal states, but pose different exploration challenges.

***Reward Density:*** Extremely spare reward structures, where the reward is zero unless a mode is reached, pose an important exploration problem. At the same time, low-reward non-zero sparse reward structures can also be very difficult to explore and are more common to find.

### 5.1.1 Sparse HyperGrid with zero rewards

We first consider the Sparse HyperGrid setting from (Pan et al., 2022), Figure 3[Left], where the reward is $+1$ only at the three corners and $0$ otherwise. Exploration difficulty is controlled through the length of horizon: larger values of $H$ make discovering the modes harder. We compare SA-GFN with a number of strong baselines such as GAFN (Pan et al., 2022), GFN with Trajectory Balance (Malkin et al., 2022), and RND with a single GFN architecture (Burda et al., 2018). We find that across all grid sizes, SA-GFN substantially outperforms all baseline methods in terms of faster convergence, efficient exploration, and an overall better learning of the underlying reward distribution as measured by the $L1$ error between the empirical distribution and the true distribution, see Figure 2.

### 5.1.2 Sparse HyperGrid with hard to explore rewards

We now consider a low-reward sparse HyperGrid with a difficult to explore reward structure. In this HyperGrid, high-rewarding modes are placed at the $2^d$ corners of a $d$-dimensional grid and relatively low (but non-zero) reward regions are located through the rest of the grid, see Figure 3[Right]. These low reward regions separating the high reward modes make mode discovery challenging.

The structure of the reward is the similar to the experiments in Malkin et al. (2022), and we consider the following two hard to explore reward settings: $(R_0 = 10^{-4}, R_1 = 1.0, R_2 = 3.0)$ and $(R_0 = 10^{-5}, R_1 = 1.0, R_2 = 3.0)$. The difficulty of exploration is controlled via reward density, grid size and action-space dimensionality. Compared to the strong baselines of GAFN (Pan et al., 2022), GFN with Trajectory Balance (Malkin et al., 2022), and RND policy with a single network architecture (Burda et al., 2018), we find that SA-GFN performs a much better exploration and a better matching of the underlying distribution, see Figure 4 and Section 10.2. Additional results are in Section 10.5.

We further visualize the distributions learnt by SA-GFN and other methods, and find that across all grid lengths, SA-GFN discovers all four modes of the distribution, while other methods are able to discover only a subset of the modes, corroborating the benefits provided by SA-GFN, Figure 5.

**Additional Experiments** on higher-dimensional HyperGrids $(d = 6, 8)$ are detailed in 10.1 and Figure 9, and for rewards setting $(R_0 = 10^{-5}, R_1 = 1.0, R_2 = 3.0)$ in 10.2. Moreover, visualiza-

tions of the learnt distributions corresponding to Figure 4 are provided in Figure 18 and Figure 19, with a further discussion in Section 10.7, to highlight the better exploration properties of SA-GFN.

## 5.2 BIT SEQUENCE GENERATION

Taken from Malkin et al. (2022), this is a auto-regressive sequence generation problem. At each time step, the policy adds a $k$-bit token to the end of the partial sequence generated so far, and the final fixed length ($n = 120$) sequence ($\mathcal{X} \in \{0,1\}^n$) is auto-regressively generated from left to right. For a fixed sequence length, by varying the values of $k$, the actual length of the trajectory ($\frac{n}{k}$) and the size of the action space or vocabulary ($|V| = 2^k$) can be efficiently controlled without changing the domain. The form of the reward function is defined in terms of a pre-defined fixed set of modes $\mathcal{M} \in \mathcal{X}$ and is completely unknown to the learning agent until it visits a terminal state and obtains the value of the reward function at that terminal state. For a given sequence $x \in \mathcal{X}$, the reward is defined in terms of the edit distance $d$ as: $R(x) = \exp(-\min_{y \in \mathcal{M}} d(x, y))$.

We evaluate the following three metrics to test the effectiveness of exploration: (a) number of modes discovered for all values of $k = \{1, 2, 4, 6, 8, 10\}$, Figure 6[Top], (b) speed of mode discovery for $k = 1$ having the longest trajectories, Figure 6[Center], and (c) the Spearman correlation between the probability of generating a sequence $p(x) = F(x)/Z$ and its reward $R(x)$ on a uniformly sampled test set, Figure 6[Bottom]. SA-GFN is compared with the baseline methods of GFlowNets with FM (Bengio et al., 2021a) and TB (Malkin et al., 2022), A2C reinforcement learning with Entropy Regularization (Williams & Peng, 1991; Mnih & Gregor, 2014) , Soft Actor-Critic (Christodoulou, 2019; Haarnoja et al., 2017), and MARS (Xie et al., 2021), an MCMC method. For all of these three metrics, we find that SA-GFN (a) discovers more modes for all values of $k$, (b) maintains a high correlation between the probability of generating a sequence $x$ and its reward $R(x)$, and (c) discovers more modes much faster when compared to other strong methods, see Figure 6, proving its robustness to the size of the action space and its effectiveness to do a better exploration.

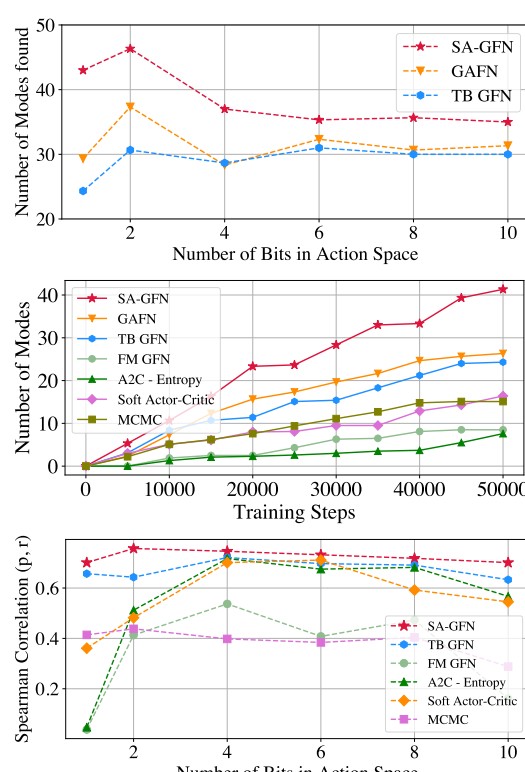

Figure 6: [**Top**] For all values of $k \in \{1, 2, 4, 6, 8, 10\}$, SA-GFN finds the highest number of modes, [**Center**] discovers more modes faster (shown for $k = 1$), [**Bottom**] has the highest Spearman correlation between the reward on a test set and the sampling probability for all $k$.

## 5.3 SMALL MOLECULE GENERATION

We further consider a more practical task of Molecule Generation from (Bengio et al., 2021a) that involves generation of binders of sEH protein (soluble epoxide hydrolase), and has a large state space of the order of $10^{16}$ and between 100 and 2000 actions depending on the state. We compare SA-GFN, with five other strong baselines: (a) GAFN (Pan et al., 2022), (b) GFN: GFlowNets trained with FM objective (Bengio et al., 2021a), (c) PPO-RND: PPO with RND rewards (Burda et al., 2018), (d) PPO (Schulman et al., 2017) and (e) MARS (Xie et al., 2021). We evaluate the (a) average reward generated by the top-10 unique molecules, (b) average Tanimoto similarity for top-10 samples, see Figure 7[Left] and Figure 7[Center], and find that SA-GFN, achieves the highest reward and has the lowest Tanimoto similarity, emphasizing high quality diverse generated candidates. We also include results for Tanimoto similarity for top-$k$ generated molecules, where

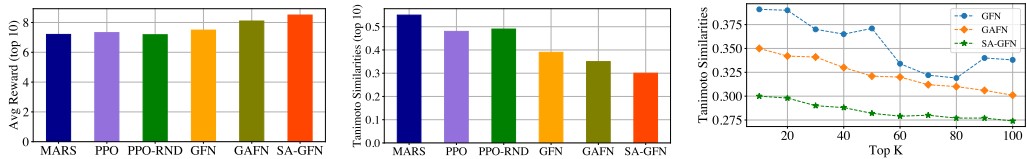

Figure 7: We find that SAGFN [**Left**] generates the highest reward, [**Center**] achieves the highest diversity in terms of Tanimoto Similarity (lower is better) of the generated molecules, and [**Right**] generates most diverse top-$k$ molecules for $k = [10, 20, 30, 40, 50, 60, 70, 80, 90, 100]$

$k = [10, 20, 30, 40, 50, 60, 70, 80, 90, 100]$ and find that SA-GFN consistently generates molecules with higher diversity, showing its better exploration properties, Figure 7[Right].

## 6 ABLATION STUDIES

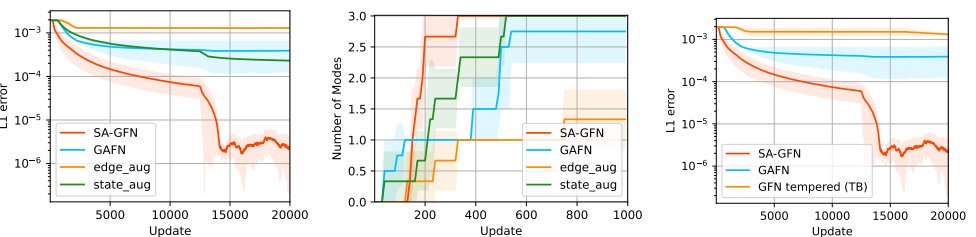

Figure 8: [**Left**] SA-GFN explores more efficiently & [**Center**] discovers modes faster as compared to state-only and egde-only intrinsic reward based single network architectures. [**Right**] Simple exploration methods, such as tempered policy logits, are not sufficient to ensure a good exploration.

**SA-GFN vs state and edge based augmentations:** We evaluated SA-GFN against a single-network variant with state & edge-level intrinsic reward augmentations for sparse reward setting and found the SA-GFN's decoupled architecture allows faster exploration and better learning of the distribution, Figure 8[Left] and Figure 8 [Center].

**SA-GFN vs simpler exploration:** Simple exploration, such as tempering logits, and entropy based methods such as SAC (Haarnoja et al., 2017), do not explore well, Figure 8[Right], 10.6 and 10.5.

## 7 DISCUSSION & CONCLUSION

We introduce Sibling Augmented Generative Flow Networks, or SA-GFN, that adopts a dual network architecture to disentangle exploration from learning, and leverages off-policy learning and trajectory relabeling to learn a behavior policy from exploratory data. SA-GFN provides an efficient exploration strategy and an easy integration of intrinsic rewards with the existing GFlowNet objectives, outperforming all other strong baselines over a wide range of exploration tasks.

*Limitations and Future Work:* SA-GFN maintains two separate networks, and future works could explore a single network variant by using two-headed architecture, for example. Multiple sibling networks (or heads) could also be incorporated to further improve exploration.

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

# 8 APPENDIX

# 9 EXPERIMENTAL DETAILS

## 9.1 EXPERIMENTAL DETAILS: SPARSE REWARD HYPERGRID WITH ZERO REWARDS

The Sparse reward HyperGrid setting consists of reward of $+1$ at the three corners of the grid (except the starting corner), and a reward of $0$ everywhere else, as visualized in Figure 3[Left]. We use increasing grid sizes with $H \in \{8, 32, 64, 128\}$. The reward function parameters and architecture choices are based on the published codebase of Malkin et al. (2022) and Pan et al. (2022). For both Sibling Network and Behavior Network, a shared network architecture with multiple heads to parameterize the GFlowNet parameters $P_F, P_B, F$ is used.

All models are trained with Adam optimizer with a batch size of 16 for a total of 20000 updates and 3 seeds. The learning rate is chosen from $\{0.001, 0.005, 0.01, 0.03\}$ for the forward and backward policies $P_F$ and $P_B$ with the trajectory balance objective (Malkin et al., 2022), and the learning rate of the $Z_\theta$ is $10\times$ learning rates of $P_F$ and $P_B$. Reward temperatures values of $\{\beta_{e_{BN}} = 1.0, \beta_{e_{SN}} = 0.25, \beta_{SN} = 1.0, \beta_{BN} = 1.0, \beta_i = 1.0\}$ are used. For intrinsic rewards, we choose RND rewards with the intrinsic reward coefficient chosen from $\{0.00005, 0.00001, 0.0005, 0.0001, 0.005, 0.001, 0.05, 0.01\}$

## 9.2 EXPERIMENTAL DETAILS: SPARSE HYPERGRID WITH DIFFICULT TO EXPLORE REWARDS

This version of the Sparse HyperGrid has a reward distribution as shown in Figure 3[Right], such that high reward "regions" are located at the $2^d$ corners of a $d$-dimensional grid, with low-reward regions everywhere else. We run all experiments on horizon sizes from $H \in \{8, 16, 32, 48, 64, 80, 96, 112, 128, 136, 144, 152\}$. The reward structure is defined as in Malkin et al. (2022) and reward function parameters of $\{R_0 = 1e^{-4}, R_1 = 1.0, R_2 = 3.0\}$. For both Sibling Network and Behavior Network, a shared network architecture with multiple heads to parameterize the GFlowNet parameters $P_F, P_B, F$ is used, and is based on the published codebase of (Malkin et al., 2022).

All models are trained with Adam optimizer with a batch size of 16 for a total of $10^6$ trajectories (62500 batches) and 3 seeds. The learning rate is chosen from $\{0.001, 0.005, 0.01, 0.03\}$ for the forward and backward policies $P_F$ and $P_B$ with the trajectory balance objective (Malkin et al., 2022), and the learning rate of the $Z_\theta$ is $10\times$ learning rates of $P_F$ and $P_B$. Reward temperatures values of $\{\beta_{e_{BN}} = 1.0, \beta_{e_{SN}} = 0.25, \beta_{SN} = 1.0, \beta_{BN} = 1.0, \beta_i = 1.0\}$ are used. For intrinsic rewards, we choose RND rewards with the intrinsic reward coefficient chosen from $\{0.00005, 0.00001, 0.0005, 0.0001, 0.005, 0.001, 0.05, 0.01\}$.

## 9.3 EXPERIMENTAL DETAILS: BIT SEQUENCES

For this task, a maximum sequence length of 120 is considered for $k \in \{1, 2, 4, 6, 8, 10\}$ and the respective vocabulary size of $2^k$. All methods use a Transformer based architecture (Vaswani et al., 2017), with 3 layers, 64 dimension, and 8 attention heads and the definition of modes $\mathcal{M}$, set of test sequences, distance metric are the same as in (Malkin et al., 2022). Reward temperatures values of $\{\beta_{e_{BN}} = 1.0, \beta_{e_{SN}} = 1.0, \beta_{SN} = 3.0, \beta_{BN} = 3.0, \beta_i = 1.0\}$ are used. Adam optimizer is used for all methods with a batch size of 16 and 50000 iterations over 3 seeds. The learning rate is chosen from $\{0.001, 0.005, 0.01, 0.03\}$ for all model parameters (except for $Z_\theta$), and the learning rate for $Z_\theta$ is chosen to be $10\times$ the learning rate of all other parameters.

## 9.4 EXPERIMENTAL DETAILS: SMALL MOLECULE

All the experiments expand on the published code of Bengio et al. (2021a) and Malkin et al. (2022). All models are trained for a maximum of 50000 batches of 4 trajectories each. The proxy model giving the reward, the held-out set of molecules used to compute the correlation metric, and the GFlowNet model architecture and its hyperparameters are taken from Bengio et al. (2021a) and Malkin et al. (2022).

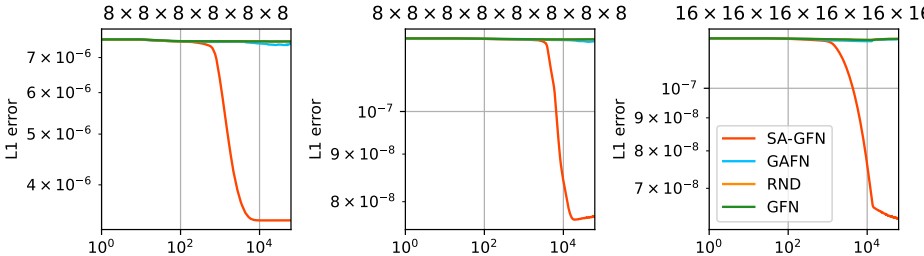

Figure 9: SA-GFN performs a better exploration and learns the underlying distribution much faster when compared to other methods even in high-dimensional HyperGrids ($d = 6, 8$) with high-dimension action spaces.

## 10    ADDITIONAL EXPERIMENTS

### 10.1    HYPERGRIDS SPANNING HIGHER DIMENSIONS

In order to test exploration capabilities of SA-GFN in the case of high-dimensional action spaces and multi-dimensional HyperGrids, we conducted experiments on the Sparse HyperGrid from Section 5.1.2 and use the same baselines as reported in Sections 5.1.1 and 5.1.2.

We found that SA-GFN outperforms all other methods even when the HyerGrid is extended to multiple dimensions and the dimensionality of the action space is increased, Figure 9. We can see that SA-GFN explores and matches the underlying distribution much better than the other baselines.

### 10.2    ADDITIONAL REWARD SETTINGS (HYPERGRID)

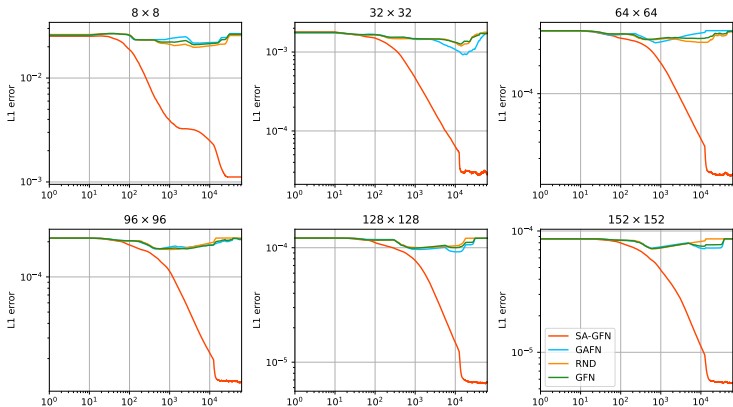

Figure 10: **Harder-to-Explore HyperGrid Setting:**   For an even harder to explore grid configuration $\{R_0 = 1e-5, R_1 = 1.0, R_2 = 3.0\}$, the proposed method, SA-GFN, outperforms all other strong baselines, highlighting its efficient exploration properties over a wide range of hard-to-explore tasks and reward structures.

To evaluate exploration over a wide range of reward structures, we include an additional HyperGrid reward setting with $\{R_0 = 1e-5, R_1 = 1.0, R_2 = 3.0\}$ that presents a more challenging exploration problem. We find that the proposed method, SA-GFN, again outperforms all other strong baselines, see 10.2, and achieves a much better match to the underlying true distribution.

### 10.3    EASY ADAPTABILITY TO OTHER INTRINSIC REWARDS

In order to highlight the versatility and ease of adaptability of the proposed method over intrinsic rewards other than regular RND (Burda et al., 2018), we add Noveld (Zhang et al., 2021) as an

intrinsic bonus to encourage exploration in the Sibling Network. The proposed method SA-GFN outperforms all other baseline methods, for both non-zero sparse reward setting, see Figure 11 and zero sparse reward settings, see Figure 12, showcasing its inclusiveness in terms of different types of intrinsic rewards.

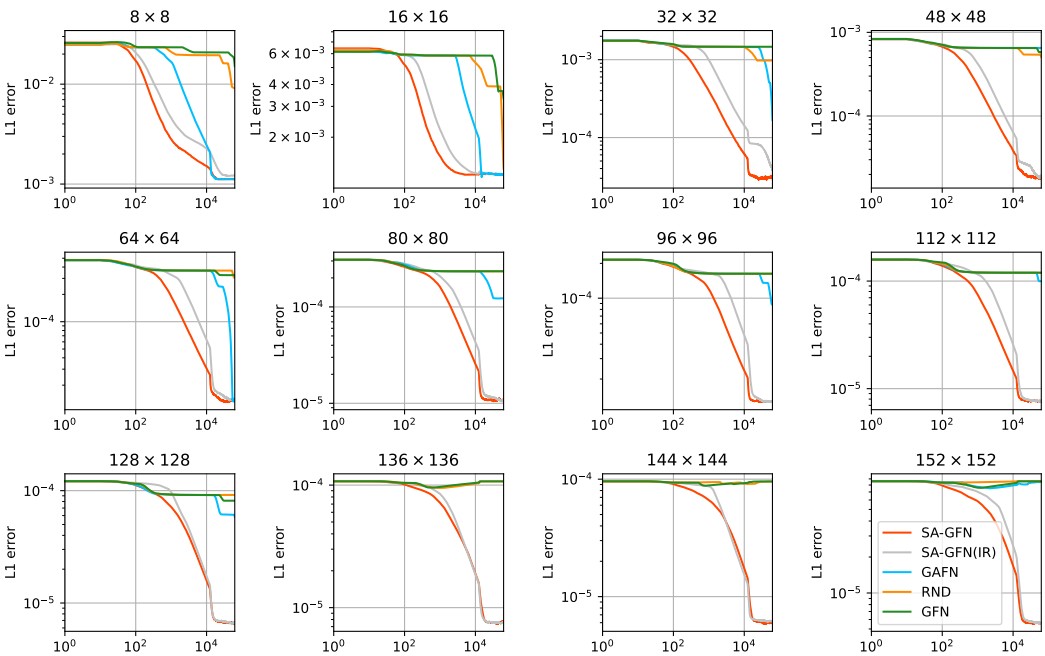

Figure 11: **Adaptability to Different Intrinsic Rewards: (non-zero sparse reward)** The proposed method, SA-GFN, is adaptable to a wide variety of intrinsic rewards. For example, SA-GFN when used with Noveld, labeled as SA-GFN(IR), outperforms all other baseline methods, achieving a much better exploration of the underlying spare non-zero reward distribution.

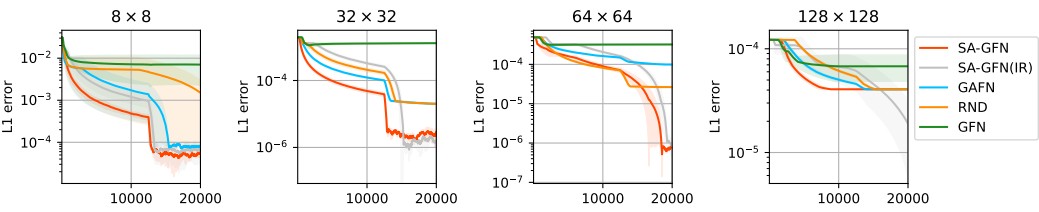

Figure 12: **Adaptability to Different Intrinsic Rewards: (zero sparse reward)** The proposed method, SA-GFN, is adaptable to a wide variety of intrinsic rewards. For example, SA-GFN when used with Noveld, labeled as SA-GFN(IR), outperforms all other baseline methods, achieving a much better exploration of the underlying sparse zero-reward distribution.

## 10.4 EXTENSION TO OTHER GFLOWNET TRAINING OBJECTIVES

The proposed method, SA-GFN can be used with any of the training objectives introduced in Section 3.2. Results using Trajectory Balance (Malkin et al., 2022) are provided in Section 5.1 and Section 5.2, while Flow Matching objective (Bengio et al., 2021a) is used in Section 5.3. Here, we include results from the Detailed Balance objective (Bengio et al., 2021b) for completeness, see Figure 13.

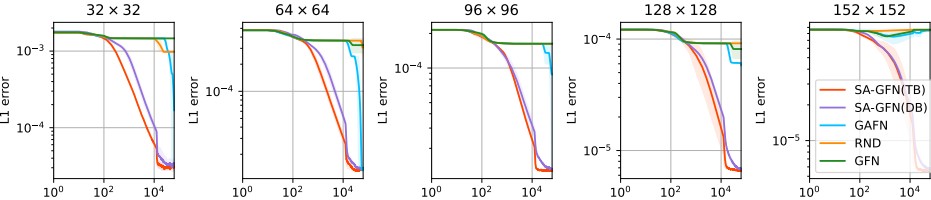

Figure 13: **Extension to other GFlowNet Training Objectives:** The proposed method, SA-GFN, can be trained using any of the GFlowNet training objectives. Here we include the Detailed Balance objective for sparse non-zero reward setting for completeness.

## 10.5 COMPARISON WITH SAC (HYPERGRID)

We add the SAC (Haarnoja et al., 2018) baseline for HyperGrid from Section 5.1 for completeness, see Figure 14 and Figure 15. We find that without adding a large replay buffer, SAC is not able to explore over difficult and sparse reward settings, while the proposed method SA-GFN performs stronger than any of the previous baselines across all reward configurations of the HyperGrid.

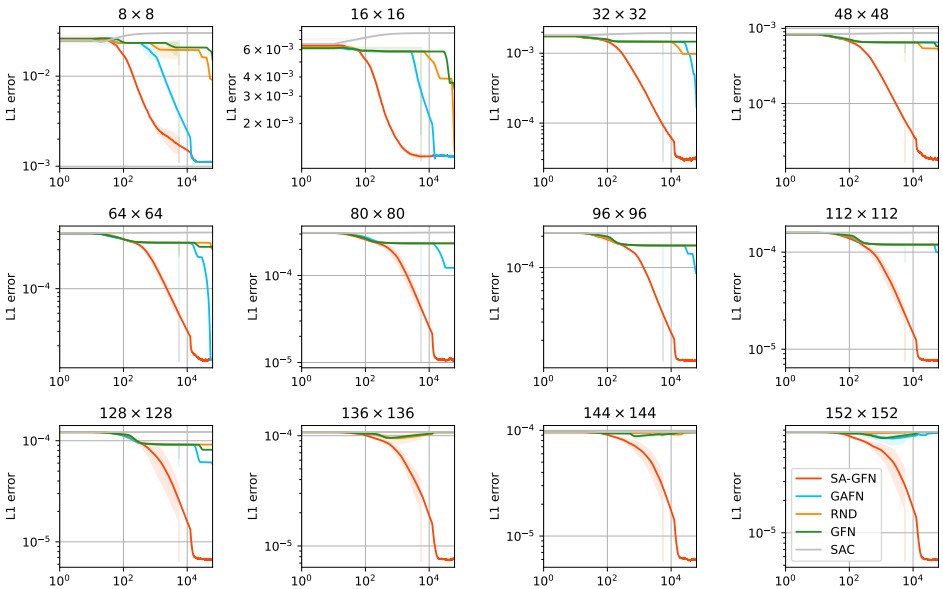

Figure 14: **Soft Actor Critic (SAC)** method is not able to explore well in difficult sparse non-zero reward settings, while the proposed method SA-GFN explores much better as compared to all other baseline methods.

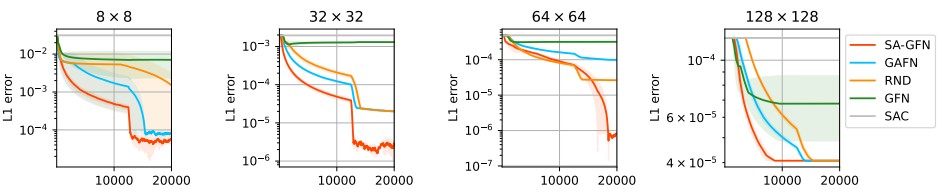

Figure 15: **Soft Actor Critic (SAC)** method is not able to explore well in difficult sparse zero-reward settings, while the proposed method SA-GFN explores much better as compared to all other baseline methods.

## 10.6 EXPLORATION USING TEMPERED LOGITS

We expand on our ablation study fom Section 6 and provide additional results to show that just using tempered logits for exploration is not sufficient for sparse and difficult to explore reward settings, see Figure 16 for sparse reward structures and Figure 17 for low-reward structures. Better exploration strategies, such as that in the proposed method, SA-GFN, become essential to ensure an efficient exploration of complex spaces and reward structures.

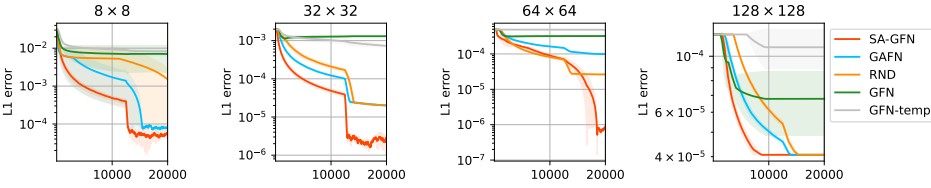

Figure 16: **Tempered logits** are not sufficient to effectively explore difficult zero-reward structures

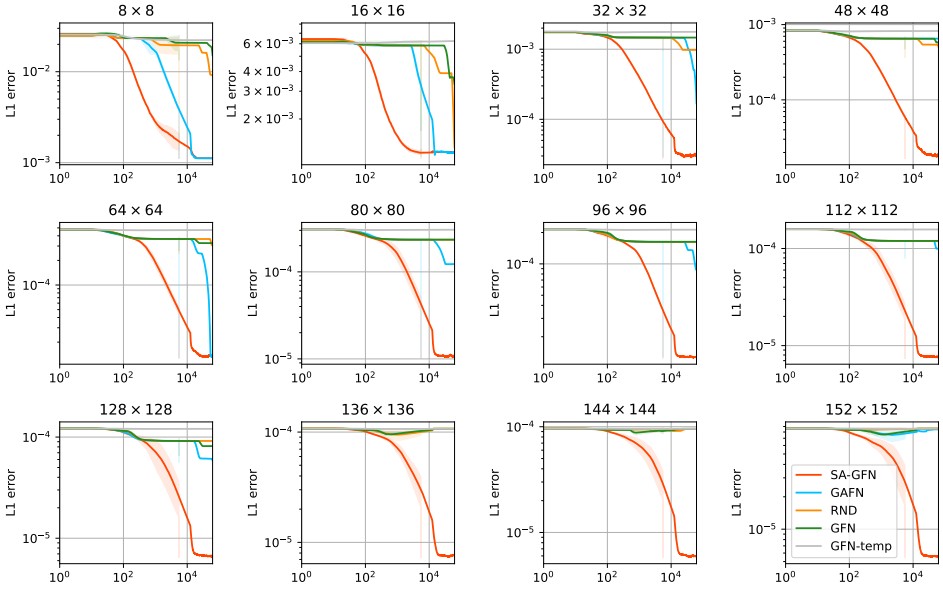

Figure 17: **Tempered logits** are not sufficient to effectively explore difficult to explore non-zero reward structures

## 10.7 VISUALIZING LEARNT DISTRIBUTIONS FOR HYPERGRID

In Figure 18 and Figure 19, we further visualize the learnt empirical distributions by SA-GFN and all other baseline methods presented in Section 5.1.2 and Figure 4.

We find that SA-GFN discovers all the four modes of the distribution for all HyperGrid sizes, including very large grid sizes, while other baseline methods suffer and are able to discover only a limited number modes as grid size increases and the exploration problem gets harder. Moreover, these visualizations confirm that SA-GFN not only discovers all the modes, but also learns and matches the true distribution well.

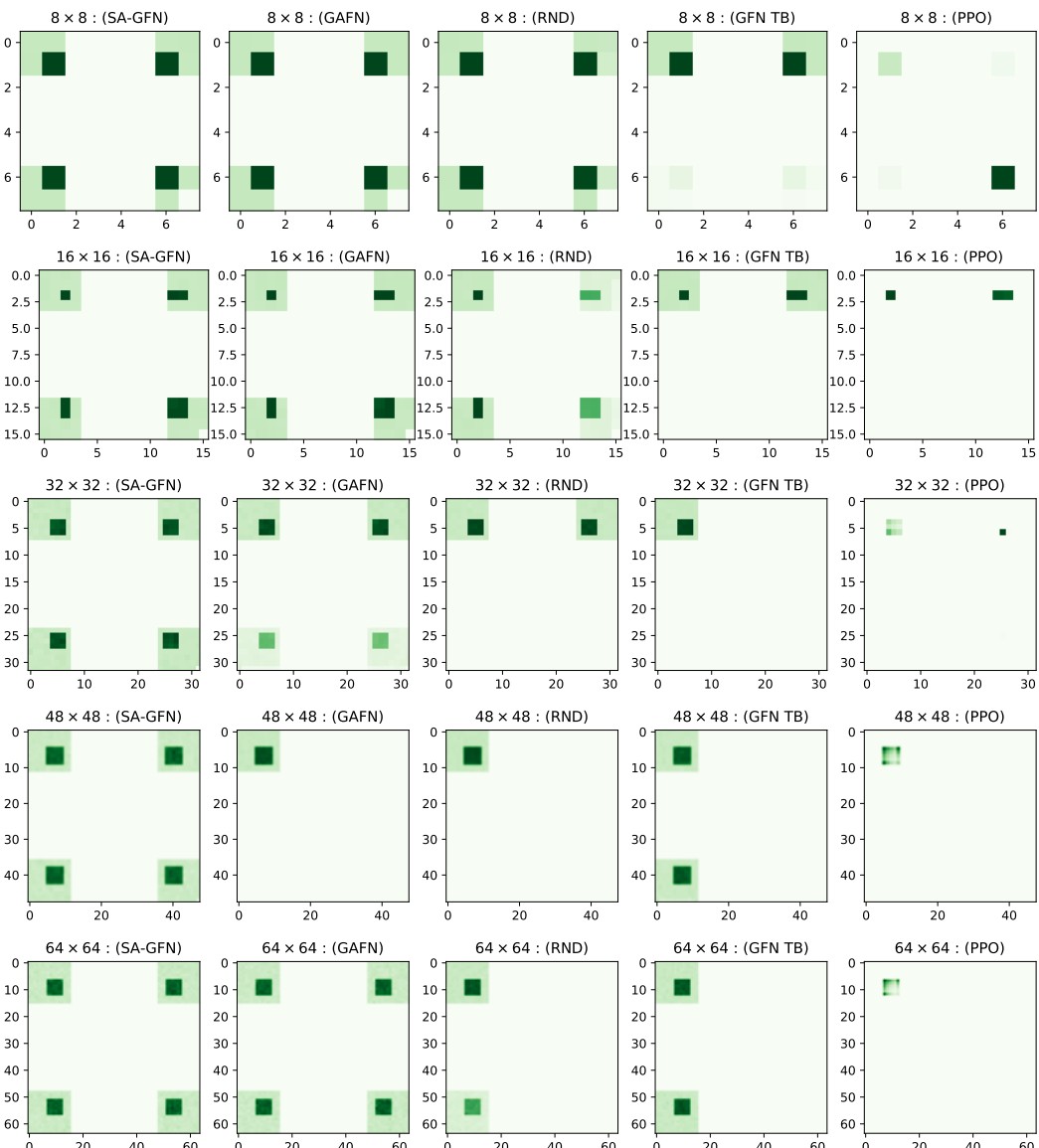

Figure 18: Visualization of the learnt empirical distributions for SA-GFN and all other methods corresponding to Figure 4. We can see that as the size of the HyperGrid increases, SA-GFN is able to discover all the four modes and learns the underlying distribution well, while other methods tend to suffer, especially as the exploration problem gets harder with increasing grid sizes.

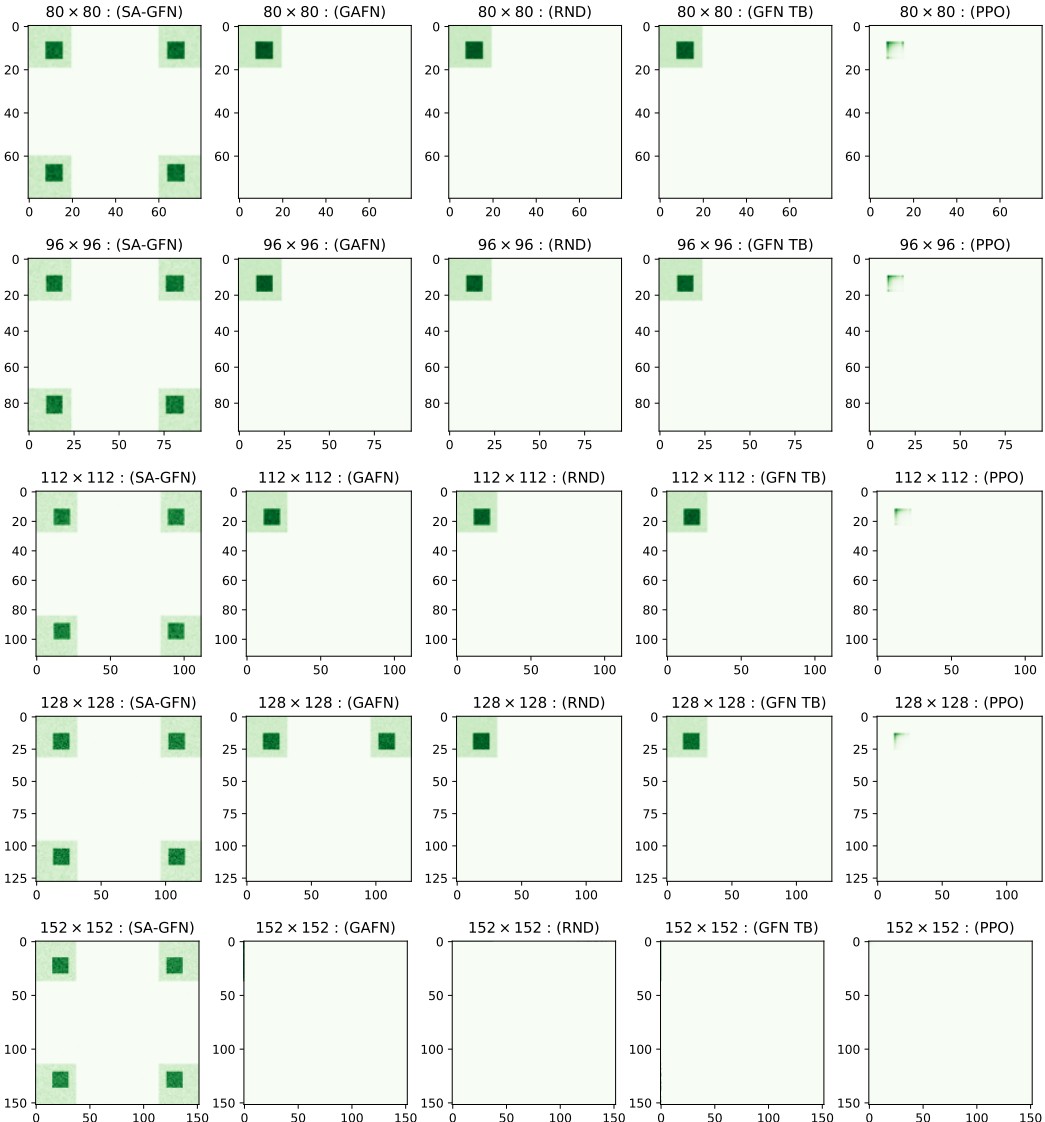

Figure 19: Visualization of the learnt empirical distributions for SA-GFN and all other methods corresponding to Figure 4. We can see that as the size of the HyperGrid increases, SA-GFN is able to discover all the four modes and learns the underlying distribution well, while other methods tend to suffer, especially as the exploration problem gets harder with increasing grid sizes.

