# OpenReview forum: "Towards Improving Exploration through Sibling Augmented GFlowNets"
_ICLR.cc/2025/Conference — ICLR 2025 Poster_

### Official Review · Reviewer_t2LW · 2024-10-26

**Soundness:** 3
**Presentation:** 3
**Contribution:** 3
**Rating:** 6
**Confidence:** 5

**Summary:**

Sibling Augmented Generative Flow Networks are proposed to solve the exploration problem in sparse reward reinforcement learning. This approach decouples the exploration policy from the exploitation policy: the exploitation policy is responsible for maximizing the cumulative extrinsic reward (enlarging the trajectory of the generation of the labeled exploration policy at the data level), and the exploration policy is responsible for maximizing the combination of the cumulative intrinsic and extrinsic reward. The authors claim that this can enhance the performance and stability of policy learning. A series of experiments verify the proposed method.

**Strengths:**

1. This paper is well-written and easy to follow.
2. The sibling policy and the algorithm is new but can be quite straightforward to come up with.
3. The experiments part provides a wide range of comparisons with existing methods.

**Weaknesses:**

1. The experimental environment is too monotonous. There is only a grid world, and there are many other environments in the sparse rewards-related research, such as Montezuma's Revenge and some robot manipulation simulation tasks in Mujoco. I am curious to see if SA-GFN would be as effective for other tasks. If the author thinks it is difficult to provide corresponding experiments, I hope there will be a qualitative analysis or discussion.
2. If I understand correctly, the author seems to think that the GFlowNets presented in this paper do not fall under the category of reinforcement learning, which I have reservations about. The goal of GFlowNets is also to maximize the cumulative discount reward, and the training paradigm of off-policy RL is also used. This is the category of reinforcement learning in my opinion. I hope the author can further explain the positioning of the method proposed in this paper.
3. In the training of SA-GFN, the critic network is not used. In general, without the critic network, policy training will be unstable because of the high variance. This paper does not provide related discussion or experiments on this aspect. I would like to hear the author's thoughts on this aspect.

**Questions:**

See weaknesses.

---

> ### Author Response · Authors · 2024-11-15
>
> We thank the reviewer for reviewing our work and providing helpful feedback. We are excited that they found our work well written, easy to follow and novel, as well as found our experiments and included baselines useful. We provide further clarifications below.
>
> - **Experimental environments:** In order to address difficult exploration settings, we include 4 hard tasks in this work: (a) Hypergrid with zero-reward, (b) Hypergrid with very low difficult to explore rewards, (c ) auto-regressive task of bit sequence generation, and (d) molecule generation with a large state space of the order of $10^{16}$ and between 100 and 2000 actions depending on the state. For the Hypergrid task, we also include larger hypergrids to allow much longer trajectories than the previous works. \
> While we agree that the RL environments suggested by the reviewer are indeed difficult to explore; in this work, we do not consider any of the RL frameworks because the current paradigm of GFNs (a) does not aim to maximize returns but instead samples from a target distribution (proportional to the rewards) and (b) only operates for non-cyclic settings of a DAG. Extensions of GFNs to more standard RL settings is a separate, and interesting, future research problem.
> - **GFlowNets vs RL:** It would be fair to characterize GFlowNets as a particular kind of RL on DAG-MDPs with a number of additional assumptions, but these differences enable new (off-policy) training objectives not equivalent to existing RL methods, which is why we see the two as distinct. There are also a lot of parallels between RL and GFlowNets and topics such as exploration and off-policy learning are common to both; however, GFlowNets mainly focus on turning an energy function into a sampler or a generative model and they do so by sampling one action at a time through a forward policy. The core objective of GFlowNets is to match a reward **distribution** (not maximize it) and turn it into a sampler, while RL mainly focuses on expected reward maximization.
> Even if GFN is within the broad family of RL and it is true that GFN methods take inspiration from RL across multiple fronts, it is important to understand this distinction and especially the fact that we are not trying to achieve the same objective. The closest RL cousin to GFN is entropy-regularized RL, and their objective coincides in some settings, where there is only one path to a given state to form a tree and not a graph, which is generally not the case in many problems.
> - **Using a Critic:** While it is possible to use a critic or a target network to our work and to GFlowNets in general, we did not use it to provide a fair comparison with other baseline methods and to mainly focus on the core problem of exploration and intrinsic rewards in this work. Additionally, while this double parameterization can help, GFlowNets can usually be optimized in a stable manner.
>
> We hope this helps in clarifying any questions the reviewer might have. We are happy to provide further clarification to any other pending concerns and suggestions and to further improve our work.

---

> > ### Author Response · Authors · 2024-12-02
> >
> > Dear Reviewer, we would like to thank you again for your time and for providing a constructive feedback for our work.
> > We have tried our best to address your concerns and questions through our rebuttal response and we hope it helped. We would appreciate any further feedback that you might have for the evaluation of our work.

---

### Official Review · Reviewer_kk92 · 2024-11-01

**Soundness:** 3
**Presentation:** 3
**Contribution:** 2
**Rating:** 6
**Confidence:** 3

**Summary:**

This paper introduces a novel architecture named Sibling Augmented Generative Flow Networks (SA-GFN), to improve the exploration capabilities in Generative Flow Networks (GFlowNets). The authors propose using a dual-network architecture, which consists of a Behaviour Network and a exploratory network (Sibling Network), which uses intrinsic rewards. By decoupling the two networks, it allows the Sibling Network to perform exploratory tasks, providing diverse data samples by relabelling the rewards with the true ones to the Behavior Network. With this design, the training objective of the behaviour network is stationary. Results shown in the selected datasets that SA-GFN outperforms baselines.

**Strengths:**

1. The paper is well written and the core idea is presented in a very clear way.
2. The idea itself is simple yet helpful for combining various intrinsic rewards methods and GFlowNets.
3. In the selected dataset, the proposed methods achieve the best performance compared to all baselines.
4. A detailed ablation study has also been provided.

**Weaknesses:**

1. Intrinsic reward method used as the baseline covers only RND. Other intrinsic reward methods such as NovelD, DEIR and others, which are shown better performance compared to RND are not included. In the appendix, NovelD is used as intrinsic reward for the sibling network but itself is still missing as a baseline
2. The tested environment is relatively simple. Testing in more complicated/hard-exploration tasks would be desirable.
3. The training of the Sibling network is unclear to me. See question section
4. An ablation study of the hyperparameter tuning of SN-GFN would be great. See question 3

**Questions:**

1. How does SA-GFN compared with other SOTA intrinsic methods, such as NovelD  then?
2. Is the sibling network also periodically updated using the data collected by the behaviour policy or directly using the behaviour policy's weight? If not, does this mean, by only trianing on the data collected by the RND, the sibling network is already sufficiently exploring the environment?
3. Currently, the training frequency of the sibling network and behaviour network seems to be 1:1. How does this ratio influence the training?

---

> ### Author Response · Authors · 2024-11-15
>
> We thank the reviewer for their feedback and review of our paper. We are excited that they found our paper well written, clearly presented, and novel. Please find answers to the comments and questions below.
>
> - **Other intrinsic rewards:** We agree that a variety of intrinsic rewards exist; what we are set out to show through this work though is that training _two_ policies, one to explore, one to behave, is advantageous in a GFlowNet setting. One can imagine then choosing their intrinsic reward of choice. RND is a simple yet effective choice as an intrinsic reward, and other methods, such as NovelD, are based on it. We include NovelD in Sec 10.3 and Fig 12 to show support for other intrinsic rewards. Further improvements and design of different types of intrinsic rewards and investigating their benefits for GFlowNets, are interesting research questions and would be a useful extension of our work.
> - **Hard exploration environments:** To evaluate on hard exploration tasks, we consider 4 difficult-to-explore settings in this work. Specifically, we (a) expanded the reward configuration in the hypergrid environment to include very low and zero reward regions to separate out modes to make exploration more difficult, (b) added the molecule generation task in Sec 5.3 that has a large state space of the order of $10^{16}$ and between 100 and 2000 actions depending on the state and is closer to a practical & hard exploration problem, (c ) evaluated on the auto-regressive bit sequence task that is still an unsolved task in terms of finding modes by the existing methods. These settings are considerably hard to explore settings, as also evident by the failure of the existing baseline methods, while our method shows considerable improvements in terms of exploration in all of them.
> - **Hyperparameter ablation:** The  main hyperparameter for our method is the weighing of the intrinsic reward term, as in RL. We keep all other hyperparameters pretty much constant across our experiments. However, we thank the reviewer for this suggestion and will consider adding a discussion around this intrinsic reward weighting hyperparameter to the final version of the paper for completeness.
> - **SA-GFN and other SOTA methods such as NovelD:** The proposed SA-GFN integrates seamlessly with NovelD through a simple plug and play in the term $r^i$ for intrinsic reward in Eq 2. For this reason, NovelD is complementary to SA-GFN and can be incorporated into SA-GFN as shown in Sec 10.3 and Fig 12. We simply chose RND through all of our experiments due to its simplicity, among other reasons mentioned in our first comment above on “other intrinsic rewards”.
> - **Sibling network training:** The Sibling Network does not use the data from Behavior Network and its main purpose is to collect exploratory data which is then relabeled and sent to the Behavior Network. Sibling network is indeed doing exploration, but is not sufficient by itself as can be seen from the “RND” baseline in which a single network using RND rewards does not learn that well.
> - **Training frequency of Sibling Network and Behavior Network:**  We indeed use a 1:1 ratio, and haven’t explored alternatives, but note that because of a dual network architecture, the training frequency of the two networks can be made completely independent and asynchronous of each other. In fact, doing so could enable an efficient parallelized training of the SA-GFN. For example, the exploratory data collected by the Sibling Network (SN) can simply be collected in a replay buffer and asynchronously sent to the Behavior Network (BN) for its training. We don’t use replay buffers in this work to provide a fair comparison with baselines, but using one is possible and can further help with efficiency.
>
> We hope this helps. We are happy to answer any other pending questions from the reviewer and to improve our work.

---

> > ### Comment · Reviewer_kk92 · 2024-11-24
> >
> > I would like to thank the authors' response which help me to have a better understanding of the paper. I'd like to maintain my score for the submission.

---

> > > ### Author Response · Authors · 2024-12-02
> > >
> > > Dear Reviewer, we thank you for your time and appreciate the positive feedback. We are glad that our response addressed your concerns.

---

### Official Review · Reviewer_GhQ4 · 2024-11-04

**Soundness:** 4
**Presentation:** 3
**Contribution:** 3
**Rating:** 8
**Confidence:** 3

**Summary:**

This work proposes a decoupled GFlowNet (Sibling) parallel with the main GFlowNet (Behavior) to improve the exploration of the learning process with sparse rewards. The auxiliary GFlowNet is trained on the original rewards combined with novelty-based intrinsic rewards, while the main network is trained only on the original rewards. Due to GFlowNet's off-policy property, the main network can use both its on-policy data and the data sampled using the sibling network. The proposed method achieves strong results in the tasks considered.

**Strengths:**

The paper is well-structured and written clearly. The storyline from the original GFlowNet through GAFN with additional intrinsic reward to SA-GFN with a decoupled network is clear and makes a lot of sense. The decoupled networks and relabeled rewards are simple yet effective methods. It is interesting to see the performance improvement by using the trajectories sampled with novelty exploration while only using the original rewards. Although intrinsic rewards are not a new idea for reinforcement learning, decoupling its samples and rewards for training has rarely been explored. The experiments are comprehensive.

**Weaknesses:**

- In the introduction, the authors claim to “expand the set of previous exploration benchmarks to include non-zero…” I assume they mean the experiment environment in section 5.1.2. It seems unfair to say so because the HyperGrid with non-zero rewards has already been used in previous work [1].
- Many references are incomplete, without information like the publisher.
- For the experiment results in Fig. 4, it would be better to plot the curves for more steps so that we can see the plateaued values of baselines, such as in 32x32, 64x64, 96x96, and so on.
- Last paragraph of 5.2. The indices of the three metrics and findings for the metrics are the same (a,b,c), which makes reading difficult.

[1] Emmanuel Bengio, Moksh Jain, Maksym Korablyov, Doina Precup, and Yoshua Bengio. Flow network based generative models for non-iterative diverse candidate generation. Neural Information Processing Systems (NeurIPS), 2021.

**Questions:**

- How is the intrinsic reward calculated? According to equation 2, the intrinsic reward depends on the terminal states and all intermediate states. Would it be different if it is only calculated for the terminal states?
- Why do some L1 error curves of baselines increase in Fig. 4, such as 8x8x8x8? I have not observed this in other related work. Please explain.
- Out of my own curiosity, do the authors see potential methods for using the decoupled intrinsic reward network in RL tasks, such as navigation?

---

> ### Author Response · Authors · 2024-11-15
>
> We thank the reviewer for their insightful review of our work. We are enthused that they found our work to be well motivated and clearly presented as well as found the experiments to be comprehensive.
>
> We provide further clarifications below.
> - **Reward setting from Section 5.1.2:** While we take the hypergrid domain from previous works [1, 2], we expand on it by including a more difficult to explore reward setting to focus more on exploration in this work. To do so, we consider a very low value of the reward hyperparameter R0 in the non-zero reward setting that makes non-mode regions sparser and difficult to explore. Specifically, the reward distribution configuration: R0=10−5, R1=1.0, R2=3.0, is hard to explore and has not been considered in the previous works for Hypergrid and lets us focus on the exploration problem.
> - **No of steps in Fig 4:** We chose the current values to keep it consistent with the previous works [1,2,3]. However, we will consider increasing the number of steps in the final version of the paper.
> - **Calculation of intrinsic reward:** The intrinsic reward calculation is exactly the same as done in RL by taking a predictor network and a fixed target network such that the representations from the target network are distilled into the predictor network.  \
> While we consider the intrinsic rewards on all states, the framework can also work with only terminal intrinsic rewards. When only terminal intrinsic rewards are considered, the second term for intrinsic rewards (in Eq 2) will now have a relatively lower value (as compared to the current form in which we add up the rewards through all timesteps). Therefore, an appropriate change in the hyperparameter $\beta_i$ will allow providing the right weighting and preference to the intrinsic rewards to encourage exploration. \
> In this work, one of the reasons for our choice of using intrinsic rewards from all timesteps is to reward the policy on taking novel paths to a given object creation. This is especially useful for GFlowNets that support a DAG structure and multiple ways to generate an object.
> - **L1-error plots in Fig 4:** Harder hypergrid configurations, such as a longer grid horizon, a higher dimension or a difficult reward configuration, can make exploration difficult. As we can see in Fig 4, as the difficulty of exploration increases (e.g. with horizon >= 136 or number of dimensions = 4), the baseline methods can sometimes diverge and/or are not able to learn in a stable manner, causing the L1 to increase. For easier configs, the behavior is much more stable.
> - **Decoupled setting in RL:** This idea of decoupling the exploration policy  network to efficiently train a second, main target policy can be used for exploration in RL as well, provided our RL policy can be trained in an off-policy manner. Specifically for navigation tasks, where both exploration and safety can be essential, an exploratory policy could be used to collect exploratory data which can then be stored, relabeled (e.g. using HER) and sent to the main RL policy that can safely learn from this data.
> - **Other edits:** We will make sure to correct the incomplete references and the indices of metrics in 5.2 - thank you for mentioning them.
>
> We hope these help in clarifying any questions that the reviewer might have. Please let us know if there are any other comments or suggestions and we would be happy to address them and further improve our work.
>
> [1] Emmanuel Bengio, Moksh Jain, Maksym Korablyov, Doina Precup, and Yoshua Bengio. Flow network based generative models for non-iterative diverse candidate generation. \
> [2] Nikolay Malkin, Moksh Jain, Emmanuel Bengio, Chen Sun, and Yoshua Bengio. Trajectory balance: Improved credit assignment in GFlowNets. \
> [3] Kanika Madan, Jarrid Rector-Brooks, Maksym Korablyov, Emmanuel Bengio, Moksh Jain, Andrei Nica, Tom Bosc, Yoshua Bengio, and Nikolay Malkin. Learning gflownets from partial episodes for improved convergence and stability.

---

> > ### Comment · Reviewer_GhQ4 · 2024-11-27
> > **Post rebuttal**
> >
> > I thank the authors for their answers to the points I raised. They were helpful and convinced me to keep my score as is.

---

> > > ### Author Response · Authors · 2024-12-02
> > >
> > > Dear Reviewer, we thank you for your time and appreciate the positive feedback. We are glad that our response addressed your concerns.

---

### Official Review · Reviewer_XUfJ · 2024-11-05

**Soundness:** 3
**Presentation:** 4
**Contribution:** 2
**Rating:** 6
**Confidence:** 2

**Summary:**

This paper is about exploration when learning generative flow networks. This is particularly relevant in a setting with sparse extrinsic terminal rewards. The proposed methods uses second, sibling, network along the main behavior network and use an intrinsic reward scheme (random network destination) for exploration. Using the two networks, it is possible to share re-labeled training data from the sibling network which is trained with the non-stationary intrinsic reward to the main behavior network with learns with the true reward. A set of abstract experiment evaluate the claims of better learning and the results show success of the proposed method. Ablation studies are also provided.

**Strengths:**

The paper addresses a relevant and interesting problem. The choice of an intrinsic reward methods and the parallel sibling network is interesting and novel for generative flow networks. Most of the experiments are fairly abstract but relevant.

**Weaknesses:**

The related work is short and makes it hard to assess novelty and relevance. The description of generative flow networks in 3.1 is short it does not become clear what the policy does and how it interacts with states and objects. RND is far from the only intrinsic reward scheme in reinforcement learning and it would strengthen the contribution to not compare against several alternative schemes.

**Questions:**

Given that RL and generative flow networks are not exactly the same, is there a way to improve RND for the generative flow network setting?

---

> ### Author Response · Authors · 2024-11-14
>
> We thank the reviewer for their feedback and for reviewing our work. We are excited that the reviewer found our work relevant, interesting and novel. We provide further context on the questions asked below:
> - **Background on GFlowNets:** Since some of the previous works [1,2,3] provide a good background on GFlowNets and due to the page limits, we kept this part short and concise. However, in the final submission, we will use an appendix to expand  the background section to make the document self-contained for the readers.
> - **RND and alternative schemes:** We agree several other methods exist, and in this work, we mainly used RND because it is a simple, yet a very competitive method for exploration. In addition, to show compatibility with the other alternatives, we provide comparison with an alternative scheme, Noveld [4] in Section 10.3 and Fig 12 in the Appendix. Our setup can also use other forms of exploration rewards, ranging from different types of pseudo counts and other novelty based methods, by simply using them in the intrinsic reward term, $r^i$, in Eq 2 of the Sibling Network.
> - **RND for GFlowNets:**  In our work, we show that more powerful intrinsic rewards (with a separate network to learn about novelty), and in particular RND, can help GFlowNets, thus confirming the hypothesis that investing compute in exploration can be a crucial ingredient for GFlowNets to realize their potential, i.e., discover yet unseen modes of the target distribution. Further improvements are an interesting research direction and can be considered an extension of this work, but would we have to speculate, we suspect that methods that prioritize modal exploration over coverage would benefit GFlowNets more - this is because GFlowNNets are set up to capture modes and can discover new ones through the generalization property of neural networks, whereas a covering/uniform exploration method might just waste some time within a mode.
>
>
> [1] Emmanuel Bengio, Moksh Jain, Maksym Korablyov, Doina Precup, and Yoshua Bengio. Flow network based generative models for non-iterative diverse candidate generation. \
> [2] Yoshua Bengio, Salem Lahlou, Tristan Deleu, Edward J Hu, Mo Tiwari, and Emmanuel Bengio. Gflownet foundations. \
> [3] Nikolay Malkin, Moksh Jain, Emmanuel Bengio, Chen Sun, and Yoshua Bengio. Trajectory balance: Improved credit assignment in GFlowNets. \
> [4] Tianjun Zhang, Huazhe Xu, Xiaolong Wang, Yi Wu, Kurt Keutzer, Joseph E Gonzalez, and Yuan-dong Tian. Noveld: A simple yet effective exploration criterion

---

> > ### Comment · Reviewer_XUfJ · 2024-11-15
> > **Thanks for the response.**
> >
> > Thanks for the response. Given the more complete explanation in the appendix, I updated the score.

---

> > > ### Author Response · Authors · 2024-12-02
> > >
> > > Dear Reviewer, we thank you for your time and appreciate the positive feedback. We are glad that our response addressed your concerns.

---

### Author Response · Authors · 2024-12-03
**General response for the rebuttal**

We sincerely thank all the reviewers for their time and for providing valuable feedback.

We appreciate their positive support and are grateful for their recognition of our paper as novel $(R_{XUfJ}, R_{GhQ4}, R_{kk92}, R_{t2LW})$, well-written and clear $(R_{GhQ4}, R_{kk92}, R_{t2LW})$, relevant $(R_{XUfJ})$ and interesting $(R_{XUfJ}, R_{GhQ4})$.

We have addressed all the concerns and questions raised during the rebuttal process and we welcome any further suggestions or opinions from the reviewers.

---

### Meta-Review · Area_Chair_Ebvk · 2024-12-16

**Metareview:**

This paper introduces, Sibling Augmented Generative Flow Networks (SA-GFN), a novel architecture to improve exploration in Generative Flow Networks (GFlowNets). SA-GFN is a dual-network with a behaviour network and an exploratory network which uses intrinsic rewards. The Sibling Network performs exploratory tasks, providing diverse data samples by relabelling the rewards with the true ones to the behavior network.

Strengths
-----------
- The paper is well-written and the motivation well-explained;
- The idea is simple to implement and effective;
- Experiments show good results and comprehensive ablations.

Weaknesses
--------------
The Reviewers appreciated the idea and the paper, but expressed some concerns about the clarity of some paragraphs. All doubts have been cleared during the rebuttal.

This paper should be definitely accepted considering the unanimous positive assessment of this paper and the significance of the proposed method. I encourage the Authors to address Reviewers' comments in the final version of the paper.

**Additional Comments On Reviewer Discussion:**

The points raised by the Reviewers mostly concerned the clarity of the writing. Doubts have been expressed about how to read figures, or interpret the technicalities of the proposed method. The Authors' rebuttal has been sufficient to clarify these concerns.

---

### Decision · Program_Chairs · 2025-01-22

Accept (Poster)